# Association of Dietary Patterns and Type-2 Diabetes Mellitus in Metabolically Homogeneous Subgroups in the KORA FF4 Study

**DOI:** 10.3390/nu12061684

**Published:** 2020-06-05

**Authors:** Nina Wawro, Giulia Pestoni, Anna Riedl, Taylor A. Breuninger, Annette Peters, Wolfgang Rathmann, Wolfgang Koenig, Cornelia Huth, Christa Meisinger, Sabine Rohrmann, Jakob Linseisen

**Affiliations:** 1Independent Research Group Clinical Epidemiology, Helmholtz Zentrum München, German Research Center for Environmental Health (GmbH), Ingolstädter Landstr. 1, 85764 Neuherberg, Germany; giulia.pestoni@uzh.ch (G.P.); anna.riedl1@gmx.de (A.R.); taylor.breuninger@helmholtz-muenchen.de (T.A.B.); c.meisinger@unika-t.de (C.M.); j.linseisen@unika-t.de (J.L.); 2Chair of Epidemiology, Ludwig-Maximilians-Universität München at UNIKA-T (Universitäres Zentrum für Gesundheitswissenschaften am Klinikum Augsburg), Neusässer Str. 47, 86156 Augsburg, Germany; 3Division of Chronic Disease Epidemiology, Epidemiology, Biostatistics and Prevention Institute, University of Zurich, Hirschengraben 84, CH-8001 Zurich, Switzerland; sabine.rohrmann@uzh.ch; 4Institute of Epidemiology, Helmholtz Zentrum München, German Research Center for Environmental Health (GmbH), Ingolstädter Landstr. 1, 85764 Neuherberg, Germany; peters@helmholtz-muenchen.de (A.P.); huth@helmholtz-muenchen.de (C.H.); 5German Center for Diabetes Research (DZD e.V.), Ingolstädter Landstr. 1, 85764 München-Neuherberg, Germany; wolfgang.rathmann@ddz.de; 6Institute for Biometrics and Epidemiology, German Diabetes Center, Leibniz Center for Diabetes Research at Heinrich Heine University Düsseldorf, Auf’m Hennekamp 65, 40225 Düsseldorf, Germany; 7DZHK (German Centre for Cardiovascular Research), Partner Site Munich Heart Alliance, Pettenkoferstr. 8a & 9, 80336 Munich, Germany; koenig@dhm.mhn.de; 8Deutsches Herzzentrum München, Technische Universität München, Lazarettstr. 36, 80636 Munich; 9Institute of Epidemiology and Medical Biometry, University of Ulm, Helmholtzstr. 22, 89081 Ulm, Germany

**Keywords:** diabetes mellitus, dietary pattern, metabotype, metabolic phenotype, Mediterranean Diet Score, Alternate Healthy Eating Index

## Abstract

There is evidence that a change in lifestyle, especially physical activity and diet, can reduce the risk of developing type-2 diabetes mellitus (T2DM). However, the response to dietary changes varies among individuals due to differences in metabolic characteristics. Therefore, we investigated the association between dietary patterns and T2DM while taking into account these differences. For 1287 participants of the population-based KORA FF4 study (Cooperative Health Research in the Region of Augsburg), we identified three metabolically-homogenous subgroups (metabotypes) using 16 clinical markers. Based on usual dietary intake data, two diet quality scores, the Mediterranean Diet Score (MDS) and the Alternate Healthy Eating Index (AHEI), were calculated. We explored the associations between T2DM and diet quality scores. Multi-variable adjusted models, including metabotype subgroup, were fitted. In addition, analyses stratified by metabotype were carried out. We found significant interaction effects between metabotype and both diet quality scores (*p* < 0.05). In the analysis stratified by metabotype, significant negative associations between T2DM and both diet quality scores were detected only in the metabolically-unfavorable homogenous subgroup (Odds Ratio (OR) = 0.62, 95% confidence interval (CI) = 0.39–0.90 for AHEI and OR = 0.60, 95% CI = 0.40–0.96 for MDS). Prospective studies taking metabotype into account are needed to confirm our results, which allow for the tailoring of dietary recommendations in the prevention of T2DM.

## 1. Introduction

During the past decades, the prevalence of type-2 diabetes mellitus (T2DM) has rapidly and dramatically increased [1]. This poses a major public health problem, especially in Western societies. It is estimated that the number of diabetic adults worldwide will increase by 50% to a total of 693 million by 2045 [2]. According to the International Diabetes Federation, about 50% of T2DM cases are undetected [2]. These numbers clearly highlight the ever-increasing severity of the diabetes epidemic and the pressing need to investigate its underlying pathophysiology so that better therapeutic strategies can be developed.

Prediabetes and diabetes mellitus are associated with an increased risk of complications, namely cardiovascular disease (CVD), retinopathy, neuropathy, and nephropathy [3]. Besides a lack of sufficient physical activity, a major role in the development of diabetes mellitus is attributed to diet [4,5]. A change in lifestyle can reduce the relative risk of developing diabetes mellitus by 40–70% in prediabetic persons [6]. Thus, the thorough investigation of dietary habits and their effects on the development of chronic diseases can have a major impact on public health.

In traditional approaches, single food groups or nutrients are examined with respect to their associations with certain diseases. Yet in reality, food groups and nutrients are not consumed in isolation. Therefore, nutritional epidemiology nowadays commonly examines dietary patterns. Dietary patterns make it possible to capture a comprehensive picture of a person’s overall diet and thereby consider possible interactions between nutrients or food groups [7,8]. A diet quality score is a dietary pattern defined a priori and is a useful tool to measure the overall quality of a diet. Two frequently used diet quality scores are the Alternate Healthy Eating Index (AHEI) [9] and the Mediterranean Diet Score (MDS) [10]. Dietary patterns have been consistently associated with T2DM in various studies investigating different ethnic populations, sexes or age groups [11,12,13,14,15,16,17,18,19,20,21,22], although metabolic characteristics have so far not been comprehensively accounted for. 

Interindividual differences in metabolism and dietary requirements are driven by intrinsic as well as extrinsic factors, e.g., (epi-)genetic background, body composition, and lifestyle [23,24,25,26,27]. The identification of metabotypes, i.e., metabolically-homogenous subgroups, has been established to adequately incorporate this knowledge when modelling diet-disease associations and in the context of personalized nutrition [28,29,30,31,32,33,34,35]. The metabotype identification consists of comparing individuals’ metabolic characteristics to each other and grouping similar individuals together. They are assigned the same metabotype. Selecting which parameters should be included in the process of defining metabotypes offers the possibility to encompass a variety of important biological processes. However, one major limitation of the concept is that there is still no unique definition of the term “metabotype” available, nor a single way to identify homogenous groups. Our group is currently investigating the selection of an optimal set of biochemical markers. A previously-published study by our group showed differential associations between single foods groups and T2DM across different metabotypes, underlining the importance of taking metabotype information into account [33]. 

The aim of the present study is therefore to examine the associations between dietary patterns and T2DM while accounting for metabotype. This is of special interest when developing targeted nutritional recommendations.

## 2. Materials and Methods 

### 2.1. Study Population

The Cooperative Health Research in the Region of Augsburg (KORA) FF4 study is a cross-sectional, population-based study that was conducted in the Bavarian region of Augsburg in 2013/2014. It is the second follow-up of the KORA S4 survey that took place from 1999 to 2001. Details on S4 as well as FF4 have been published previously [36,37]. Briefly, 2279 individuals, the vast majority with Caucasian ethnicity, participated in the KORA FF4 study. They were physically examined, answered self-administered questionnaires, and participated in computer-assisted face-to-face interviews during their study center visit. Fasting blood samples were taken at the study center and immediately underwent preprocessing. For our analyses, we excluded participants without dietary information (*n* = 677, leaving 1602), without metabotype information (*n* = 41, leaving 1561), with type 1 diabetes (*n* = 3, leaving 1558), unclear glucose tolerance status due to missing oral glucose tolerance test (OGTT) information (*n* = 38, leaving 1520), or unclear/unvalidated glucose tolerance status (*n* = 1, leaving 1519). Furthermore, participants with missing information on their hypertensive status (*n* = 2, leaving 1517), and those with severe diseases, namely myocardial infarction (*n* = 46) and/or stroke (*n* = 34) and/or cancer (*n* = 158) were also excluded. The final data set for the analysis included 1287 participants.

The KORA studies were approved by the Ethics Committee of the Bavarian Chamber of Physicians and carried out in accordance with the Declaration of Helsinki (EK Nr. 06068, 25.10.2012). Written informed consent was obtained from all participants.

### 2.2. Covariates 

The analyses described below included age (in years) and waist circumference (in cm) as continuous variables, sex (male/female), smoking status (never/former/current), education (< 10 years/10–12 years/> 12 years), physical activity (active/inactive) and actual hypertensive status (yes/no) as categorical covariates. The categorization of the education variable represents the German education system. Physically-active participants are those who were active during summer and winter and at the same time active for at least 1 h per week during at least one season. Hypertensive individuals included those with measured systolic blood pressure ≥ 140 mmHg and/or diastolic blood pressure ≥ 90 mmHg and/or use of anti-hypertensive medication given that the subjects were aware that they had hypertension.

### 2.3. Metabotypes 

Participants were grouped into three metabolically-homogenous subgroups. Details are given in References [34] and [33]. Briefly, a set of 16 clinical markers (body mass index (BMI), 12 laboratory parameters measured in serum, i.e., glucose, total cholesterol, high-density lipoprotein cholesterol, total cholesterol/high-density lipoprotein cholesterol ratio, low-density lipoprotein cholesterol, uric acid, triglycerides, gamma-glutamyltransferase, glutamate-pyruvate transaminase, glutamate-oxaloacetate transaminase, alkaline phosphatase high-sensitivity C-reactive protein and insulin, and two parameters measured in fresh venous whole EDTA blood, i.e., leukocytes and glycated hemoglobin) was used to identify and characterize three clusters by applying the k-means clustering algorithm. These 16 clinical markers are the available subset of markers originally used in the development of this metabotype concept. The prevalence and incidence of metabolic diseases in the clusters are used to quantify how well the clinical markers define distinct clusters. Missing observations were imputed by MICE (multivariate imputation by chained equations) and values were standardized prior to running the clustering algorithm. Metabotypes were derived in a larger sample of 2218 (out of 2279 total) FF4 participants. Only participants who did not fast for at least 8 h before blood collection (*n* = 54) or participants with more than 10% missing values of all clustering variables (*n* = 7) were excluded. The three cluster solution presented in Reference [33] describes cluster 3 as the most unfavorable metabotype with respect to the metabolic characteristics, e.g., the highest median concentrations of serum glucose and glycated hemoglobin, as well as other risk factors for T2DM, including age, obesity, physical inactivity, family history of diabetes and hypertension. In contrast, clusters 1 and 2 define rather beneficial metabotypes; cluster 1 shows even healthier characteristics than cluster 2 which can be considered as an “intermediate” cluster.

### 2.4. Dietary Intake and Diet Quality Scores 

Usual dietary intake was calculated based on one food frequency questionnaire (FFQ) and up to three 24 h food lists (24HFL) per participant. One 24HFL was filled in by the participants during their study center visit, and up to two further 24HFLs were filled in at home within the following three months. In contrast to that assessment of the short-term diet, the FFQ assesses the participants’ diet over the last 12 months. Details are given elsewhere [38]. Briefly, a two-step approach was used to model consumption probabilities and consumption amounts. Usual intake estimates for the food items were further categorized into food groups based on the EPIC-Soft classification system [39]. Furthermore, estimates of nutrient intake were derived for every participant by linking the usual food group intake to the National Nutrient Database (Bundeslebensmittelschlüssel 3.02).

As measures of quality of the overall diet of the participants, the Alternate Healthy Eating Index (AHEI) 2010 [9] and the Mediterranean Diet Score (MDS) 2003 [10] were calculated based on the usual intake estimates. The AHEI 2010 consists of 11 food components, namely vegetables, fruits, whole grains, sugar-sweetened beverages (SSB) and fruit juice, nuts and legumes, red/processed meat, trans fat, long-chain (n-3) fatty acids, polyunsaturated fatty acids (PUFA), sodium and alcohol. Dietary intake of fish and shellfish was used as a proxy for long-chain (n-3) fatty acid intake. Based on the quantity of each food component consumed, 0 to 10 points were attributed, yielding a theoretical range of 0–110 points for the total AHEI score. The higher the score, the healthier the diet is considered to be. Cut-off values which result in minimum and maximum points per dietary component are given, and intermediate intakes are scored proportionally between those limits. For example, the maximum score of 10 is attributed if at least 5 servings of vegetables per day were consumed. Scoring servings proportionally would, therefore, attribute 2 points for every serving, that is 4 points for 2 servings, 6 points for 3, and so on. These cut-offs have been updated in the 2010 version of the original AHEI [40] according to state-of-the-art practices in nutritional science. In our analysis, the usual intake in g/day had to be converted to servings per day by using standard portion sizes as reported in Reference [9]. For example, one serving of green leafy vegetables equals 1 cup or 236.59 g and for all other vegetables, 1 serving equals 0.5 cup or 118.295 g. Furthermore, we had to exclude the trans-fat component, as this information is not available in our data set. For this reason, the modified AHEI computed in the present study with 10 components ranged theoretically from 0 to 100 points. Table 1 summarizes the components of the modified AHEI and the cut-off values applied in the analysis.

The MDS 2003 includes 9 dietary components, namely vegetables, legumes, fruits and nuts, cereals, fish, meat, dairy products, alcohol, and fat intake (ratio of monounsaturated to saturated fatty acids), where fish has been additionally included to the index originally developed [41]. For each of the components listed above, a value of either 0 or 1 point was assigned to each participant. Except for alcohol intake, sex-specific medians were used as cut-off values. Apart from the components of meat and dairy products, participants with a usual intake above the median for a given component were assigned a score of 1 and those below a score of 0. For the components of meat and dairy products, participants above the median were assigned to a score of 0 and below to a score of 1. Alcohol intakes between 10 and 50 g per day for men and between 5 and 25 g per day for women were considered as optimal, therefore leading to a score of 1 for this component. The total MDS calculated in the present study ranged from 0 to 9 points, where 0 reflects a minimal adherence and 9 reflects a strong adherence to the traditional Mediterranean diet. Table 2 summarizes the sex-specific cut-offs applied in our study.

### 2.5. Outcome 

For participants without a diagnosis of diabetes mellitus, glucose tolerance status was determined following the criteria published by the American Diabetes Association [42] based on an OGTT. Criteria that defined normal glucose tolerance status were a fasting glucose concentration of less than 5.6 mmol/L or a 2-h OGTT concentration of less than 7.8 mmol/L. Impaired glucose tolerance (2-h OGTT concentration of 7.8–11.0 mmol/L) and/or impaired fasting glucose (fasting glucose concentration of 5.6–6.9 mmol/L) was classified as prediabetes. Those participants with a fasting glucose concentration of at least 7.0 mmol/L or a 2-h OGTT concentration of at least 11.1 mmol/L were categorized under undetected diabetes mellitus. Participants with prevalent diabetes mellitus were identified by a self-reported diagnosis of T2DM or by the use of anti-diabetic medication. Prevalent cases were validated by contacting the participants’ physicians. For the analysis, participants with prevalent and undetected diabetes mellitus were grouped as “T2DM participants” and participants with prediabetes and those with normal glucose tolerance status were grouped as “non-diabetic participants”.

### 2.6. Statistical Analysis 

The characteristics of the KORA FF4 participants are described for the total study population and stratified by metabolically-homogenous clusters. To assess associations between T2DM (dependent variable) and diet quality scores (independent variables), we fitted three different logistic regression models, where the AHEI was scaled by 10 points and the MDS was scaled by 2 points. Model 1 was adjusted for age, sex, and energy intake. Model 2 incorporated additional adjustment for education, physical activity, and smoking. Finally, Model 3 extended Model 2 by including waist circumference and hypertension status. All models were fitted to the total data set as well as stratified by metabotypes. Models fitted to the total data set were presented with and without further adjustment for metabotype. Additionally, for models fitted in the metabotype 1 cluster, smoking status was incorporated as a binary covariate (i.e., never smoker vs. current and former smokers) due to the limited sample size in this cluster. We report odds ratios (OR) and 95% confidence intervals (CI). Additionally, the interaction effects between metabotype and both diet quality scores were assessed by performing Likelihood-Ratio tests comparing the models, including the interaction between metabotype and diet quality scores with the models without interaction. 

All statistical analyses were carried out in R software (version 3.5.3 for Windows, R Foundation for Statistical Computing, Vienna, Austria.). The significance level was set to 0.05 for all results.

## 3. Results

The characteristics of the study sample are described in Table 3 in total and stratified by the three metabotypes. In the total study sample, there were slightly more women (53.4%) than men and the mean age was 58.3 ± 11.6 years. Subjects in cluster 3 were more often men (64.1%), had the highest mean age (62.0 ± 11.1 years), highest BMI (32.7 ± 5.7 kg/m^2^), the lowest proportion of current smokers (11.4%) and the highest proportion of prevalent diabetic participants (30.0%) compared to the other two metabotype clusters. Furthermore, metabotype cluster 3 was characterized by the lowest proportion of physically-active participants (43.6%) and the highest proportion of hypertensive participants (65%). Regarding dietary intake, these participants showed the highest mean consumption of meat and meat products (140.1 g/day) and the lowest mean consumption of vegetables (158.1 g/day). Subjects in cluster 1 appeared to be the healthiest and show the highest proportion of normal-weight participants, the highest proportion of physical active status, the lowest proportion of hypertension and the highest proportion of normal glucose tolerance status. Characteristics of participants in cluster 2 were intermediate to cluster 1 and cluster 3. There was little variation in the mean of the two diet quality scores between the metabotype clusters and both measures were in very good accordance, even though they incorporate some different food components. A detailed characterization of the metabotype clusters with respect to the clinical and anthropometric markers for the FF4 study is given in the Supplementary material of Riedl et al. [33].

Results from the logistic regression analyses are presented in Table 4. Without including metabotype information, significant inverse associations between T2DM and both scores were found in Models 1 and 2 for the total study sample. For both diet scores, an additional adjustment for education, physical activity, and smoking led to slightly higher OR estimates that are still below 1. The AHEI predicted a 36% reduction of T2DM odds compared to a 26% reduction predicted by the MDS. When additionally including the metabotype variable as adjusting factor in the total model, a significant inverse association of T2DM and the diet quality scores was only observed when the basic adjustment (age, sex, and energy intake) was applied. Remarkably, a significant interaction effect between diet score and metabotype was found in Model 2 (both scores) and in Model 3 (AHEI). To further examine these interaction effects, we conducted a stratified analysis. In the analysis stratified by metabotype, we found significant inverse diet-diabetes associations in cluster 3 only. It is noteworthy that these associations persisted with further adjustment. A significant 40% reduction of T2DM odds was observed for every 10-point increase in AHEI in Model 2. Likewise, a significant 38% reduction of the odds of diabetes mellitus was observed for every 2-point increase of the MDS in Model 2. When comparing the different Models to each other, we observed that further adjustment only slightly affected OR estimates. No significant associations with T2DM were seen for the diet quality scores within metabotype cluster 1 or cluster 2.

## 4. Discussion

We investigated the association between T2DM and two diet quality scores while taking into account the role of metabotypes in the population-based KORA FF4 study. We found significant interaction effects between metabotype information and both diet quality scores. In the analysis stratified by metabotype, significant inverse associations between T2DM and the two diet quality scores were only detected in the metabolically-unfavorable cluster. We observed a reduction in the odds of T2DM of about 40% for both diet scores in cluster 3. No significant associations were identified within metabotype clusters 1 and 2, i.e., the more metabolically-favorable metabotypes. Without accounting for metabotype information, significant negative associations between T2DM and both quality scores were observed in the total sample. The reduction of the odds for T2DM in the total sample was lower for both diet scores (36% and 24%) in comparison to the estimates derived from the stratified analysis in cluster 3.

When additionally including waist circumference and hypertension as adjustment variables in the logistic regression models, the association between T2DM and diet quality scores was attenuated. This might be explained by the fact that these two factors are potential mediators of diet and T2DM association (e.g., References [16,44,45,46]). It is known that mediating factors should not be included as adjusting factors in regression models. Therefore, our Model 3 was potentially over-adjusted. The cross-sectional design of the KORA FF4 study does, however, not allow for examining the potential mediation effect of the above-mentioned variables. The inverse association between the diet scores and T2DM was significant after adjustment for energy intake and physical activity. This supports the idea that diet quality is an independent predictor of T2DM status, independent of calorie deficit. 

Following a Mediterranean diet has been shown to have protective effects on the development of T2DM by increasing insulin sensitivity and decreasing oxidative stress and inflammation [47]. Anti-inflammatory mechanisms are linked to n-3 fatty acids, which are typically consumed in larger quantities on a Mediterranean diet high in fish, legumes, and nuts [48,49,50,51]. Consumption of antioxidants also has positive effects on beta cell function and insulin resistance [52]. The consumption of these dietary components is assessed by the AHEI analogously.

It has been shown that a high-quality diet, as measured by the AHEI and the MDS indices, is inversely associated with the risk for morbidity and mortality of the major chronic diseases, namely CVD, cancer and T2DM [12,14,15,17,53,54,55,56,57,58,59,60,61,62,63,64]. We examined both indices as they have different features, although they appear similar with respect to the majority of components used. In contrast to fixed cut-off values as used by the AHEI, the MDS applies sex-specific medians of the population that is studied. Therefore, a comparison between different populations or the examination of changes over time is not an appropriate use of the MDS. On the other hand, the MDS approach ensures an equal contribution of all components to the total score [7]. In our analyses, we obtained similar results for both diet scores. This consistency strengthens our findings.

### 4.1. A Priori Dietary Patterns and T2DM 

Without including metabotype information in the logistic regression models, we detected significant associations of T2DM and both diet quality scores in the total sample (OR = 0.64, 95% CI = 0.49–0.83 for AHEI, OR = 0.74, 95% CI = 0.57–0.96 for MDS). Several other studies have examined the association between AHEI and T2DM. The results are mostly in line with our findings of significant associations between T2DM and diet quality scores. The Health Professionals Follow-Up Study showed a reduced risk of T2DM with increasing AHEI and increasing alternate MDS (analogous to the MDS but no dairy intake included) in American men [12]. The absolute risk reduction was accentuated among overweight and obese participants [12]. In postmenopausal women aged 50–79 years participating in the Women’s Health Initiative, the AHEI and the alternate MDS were inversely associated with T2DM (multivariable-adjusted hazard ratios 0.87 and 0.90 per 1-SD increase in score, respectively) [15]. In the Multiethnic Cohort Study with participants aged 45–75 years, the AHEI and the alternate MDS, among other indices, were examined with respect to their association with T2DM. Both diet quality scores were inversely associated with T2DM [17]. The direction of the associations was mostly consistent across all ethnic groups [17]. The AHEI has been shown to be inversely associated with the risk of having T2DM in an Asian population aged 45–74 years (multivariable-adjusted hazard ratio 0.93 per 1-SD increase in score) [19]. A stratified analysis revealed a significant interaction effect of smoking on the association which was only present in non-smokers (p-interaction 0.03) [19]. This holds for the alternate MDS as well (multivariable-adjusted hazard ratio of 0.93 per 1-SD increase in score).

In American men and women from three prospective cohort studies (Nurses’ Health Study (NHS), NHS II and Health Professionals Follow-Up Study) that were analyzed jointly, an improvement of diet and a worsening of diet as assessed by a change in AHEI score have been shown to be associated with a lower and higher risk of T2DM, respectively [16]. Recently, the Atherosclerosis Risk in Communities (ARIC) study did not confirm an association of AHEI and the risk of diabetes mellitus in black and white American men and women aged 45–64 years [60]. However, a significant trend of risk reduction for CVD was observed across AHEI quintiles [60].

Based on 6-year follow-up data from the Multi-Ethnic Study of Atherosclerosis (MESA), no significant association between the MDS and risk of incident diabetes mellitus was found in men and women aged 45–84 years at baseline [11]. In the EPIC-Potsdam Cohort (men and women aged 35–64 years), the MDS was significantly inversely associated with risk of diabetes mellitus (hazard ratio 0.93 per 1 SD increase in score) [18]. Earlier, an analysis of data from 8 EPIC cohorts showed a significant inverse association of the relative MDS (including the same components of the original MDS but using tertiles, not the median) and risk of T2DM [44]. The results of systematic reviews and meta-analyses confirm a relevant risk reduction of diabetes mellitus when following a Mediterranean diet as assessed by the MDS [13,14].

### 4.2. Inclusion of Metabotype Information to Identify High-Risk Strata 

The inclusion of metabotype information when investigating the association of dietary patterns and T2DM makes our study unique. We only found significant associations between diet quality and T2DM in the cluster representing a metabolically unfavorable metabotype. Numerous cohort studies examined the effect of following certain dietary patterns on the risk of incident T2DM. In most studies, inverse associations were revealed, offering a promising possibility to implement prevention actions for populations based on such dietary patterns. These approaches, however, do not take into account other factors, either intrinsic or extrinsic, that influence the response or non-response to a dietary pattern. Important examples include the composition of gut microbiota and responsiveness to a Nordic diet, as discussed in a review paper [65], the relationship of weight loss and insulin and glucose status [66], or vitamin D responsiveness and markers of metabolic syndrome [67]. Our analysis included metabotype information as a novel feature when analyzing the association between T2DM and dietary patterns. Metabotypes describing metabolically-similar individuals have previously been suggested [31] and successfully applied in the field of tailored dietary advice [35]. Therefore, we expected to see differences in the associations of T2DM and dietary patterns across the metabotypes. Only in cluster 3, the metabolically least-favorable cluster, were significant inverse associations between T2DM and the diet quality scores present. Prospective studies including metabotype information are needed to further elucidate a possible causal relationship. This could confirm the hypothesis that the protective effect of diet on T2DM is modified by underlying metabotypes, i.e., by metabolic characteristics. This would allow for the identification of metabolically-homogenous subgroups that would benefit the most from the improvement of the quality of the diet as measured by AHEI and MDS. 

### 4.3. Strength and Limitations 

The KORA FF4 study is the second follow-up of KORA S4, a population-based study in the Augsburg study region. The KORA S4 survey was characterized by a high participation rate and was representative of the general Bavarian population. The assessment of glucose tolerance status was carried out by trained and experienced staff and has been validated. The dietary assessment was extensive, and the usual intake was derived by applying a sophisticated method. The estimation of usual intake is based on combining information obtained by long-term and short-term dietary assessment instruments. The application of two diet quality scores and the subsequent results obtained in our analyses strengthen our findings. Nonetheless, the number of prevalent diabetic or unknown diabetic participants is rather small. Due to the limited sample size, some categories for glucose tolerance status had to be combined for the analysis. This resulted in a loss of detailed information. A further limitation of our study is its cross-sectional nature. Our analysis does not allow for drawing causal conclusions about the association of T2DM and a priori dietary patterns. Still, we assume that including undiagnosed diabetic patients in the group of cases strengthens our findings. These participants were not aware of the diagnosis and therefore did not make healthy changes to their dietary behavior, as was probably the case for the already diagnosed participants. Regarding the dietary assessment, we cannot rule out recall bias or over- or under-reporting. One major limitation of the metabotype concept is that there is still no unique definition of the term “metabotype” available, nor a single way to identify homogenous groups, as reviewed in Reference [32] and recently in Reference [68], with a focus on nutrition.

## 5. Conclusions

We investigated the role of metabotypes in the association of dietary patterns and T2DM. The inverse association between AHEI and MDS and T2DM was only present among those with an unfavorable metabotype. Prospective studies including metabotype information are needed to confirm these results, which allow for the tailoring of dietary recommendations in the prevention of T2DM.

## Figures and Tables

**Table 1 nutrients-12-01684-t001:** Criteria for the definition of the modified Alternate Healthy Eating Index [42].

Component	Criteria for Minimum Score (0)	Criteria for Maximum Score (10)
Vegetables, servings/d	0	≥5
Fruits, servings/d	0	≥4
Whole grains, g/d		
Women	0	75
Men	0	90
SSB and fruit juice, servings/d	≥1	0
Nuts and legumes, servings/d	0	≥1
Red/processed meat, servings/d	≥1.5	0
Fish, g/d	0	≥32.4
PUFA, % of energy	≤2	≥10
Sodium, mg/d		
Women	≥3337	≤1112
Men	≥5271	≤1612
Alcohol, drinks/d		
Women	≥2.5	0.5–1.5
Men	≥3.5	0.5–2.0
Total score	0	100

Adapted from [9]. Abbreviations: SSB: sugar-sweetened beverages; PUFA: polyunsaturated fatty acids.

**Table 2 nutrients-12-01684-t002:** Criteria for the definition of the Mediterranean Diet Score [10].

Component	Criteria for Maximum Score (1)	
	Women	Men
Vegetables	≥181.50 g	≥148.50 g
Legumes	≥5.30 g	≥4.25 g
Fruits and nuts	≥161.15 g	≥156.85 g
Cereals	≥246.05 g	≥311.35 g
Fish	≥14.50 g	≥18.45 g
Meat	<85.15 g	<140.05 g
Dairy products	<199.45 g	<152.35 g
Alcohol	5–25 g	10–50 g
Fat intake (MUFA:SFA)	>0.75	>0.80

Abbreviations: MUFA: monounsaturated fatty acids, SFA: saturated fatty acids.

**Table 3 nutrients-12-01684-t003:** Characteristics of the study population overall and by metabotype.

	Overall		Cluster 1		Cluster 2		Cluster 3	
	(*n* = 1287)		(*n* = 591)		(*n* = 476)		(*n* = 220)	
Sex, *n* (%)								
Male	600	(46.6)	182	(30.8)	277	(58.2)	141	(64.1)
Female	687	(53.4)	409	(69.2)	199	(41.8)	79	(35.9)
Age (years)	58.4	(11.6)	55.9	(11.8)	59.7	(11.1)	62	(11.1)
Education, *n* (%)								
< 10 years	67	(5.2)	23	(3.9)	29	(6.1)	15	(6.8)
10–12 years	736	(57.2)	313	(53.0)	284	(59.7)	139	(63.2)
≥ 13 years	484	(37.6)	255	(43.1)	163	(34.2)	66	(30.0)
BMI (kg/m^2^)	27.5	(4.9)	24.9	(3.5)	28.3	(3.7)	32.7	(5.7)
BMI categorized, *n* (%)								
Underweight	6	(0.5)	6	(1.0)	0	(0.0)	0	(0.0)
Normal weight	422	(32.8)	323	(54.7)	89	(18.7)	10	(4.5)
Overweight	526	(40.9)	216	(36.5)	244	(51.3)	66	(30.0)
Obese	333	(25.9)	46	(7.8)	143	(30.0)	144	(65.5)
Waist circumference (cm)	95.6	(14.2)	87.4	(11.5)	98.9	(10.1)	110.9	(13.1)
Physical activity, *n* (%)								
Inactive	488	(37.9)	183	(31.0)	181	(38.0)	124	(56.4)
Active	799	(62.1)	408	(69.0)	295	(62.0)	96	(43.6)
Smoking status, *n* (%)								
Never	553	(43.0)	265	(44.8)	202	(42.4)	86	(39.1)
Current	182	(14.1)	86	(14.6)	71	(14.9)	25	(11.4)
Former	552	(42.9)	240	(40.6)	203	(42.6)	109	(49.5)
Hypertension, *n* (%)								
No	833	(64.7)	459	(77.7)	297	(62.4)	77	(35.0)
Yes	454	(35.3)	132	(22.3)	179	(37.6)	143	(65.0)
Glucose tolerance status, *n* (%)								
Normal glucose tolerance	689	(53.5)	429	(72.6)	233	(48.9)	27	(12.3)
Prediabetes	453	(35.2)	137	(23.2)	211	(44.3)	105	(47.7)
Undetected diabetes	48	(3.7)	7	(1.2)	19	(4.0)	22	(10.0)
Prevalent diabetes	97	(7.5)	18	(3.0)	13	(2.7)	66	(30.0)
Usual intake (g/day)								
Vegetables	175.9	(60.1)	190.2	(66.5)	166.5	(51.0)	158.1	(50.8)
Fruits and nuts	167.5	(85.7)	172.8	(83.4)	162.8	(86.8)	163.5	(89.0)
Meat and meat products	115.9	(43.9)	102.1	(37.0)	121.9	(43.0)	140.1	(49.4)
Fish and Shellfish	20.5	(13.1)	20.2	(13.7)	20.4	(11.8)	21.3	(14.0)
Diet quality scores								
Alternate Healthy Eating Index	44.4	(8.8)	46.1	(8.9)	43.6	(8.3)	41.5	(8.6)
Mediterranean Diet Score	4.3	(1.8)	4.5	(1.9)	4.3	(1.8)	4.1	(1.7)

Continuous variables: mean (SD), categorical variables: n (%). BMI categorized following the WHO definition (underweight: BMI < 18.5 kg/m^2^, normal weight: 18.5 ≤ BMI < 25.0 kg/m^2^, overweight: 25.0 ≤ BMI < 30 kg/m^2^, obese: BMI ≥ 30 kg/m^2^) [43]. BMI: body mass index, SD: standard deviation.

**Table 4 nutrients-12-01684-t004:** Associations Between Diet and T2DM in the Total Sample, Stratified by Metabotype.

	Type-2 Diabetes Mellitus										
	Total (*n* = 1287)		Total (*n* = 1287) ^d^			Cluster 1 (*n* = 591) ^e^		Cluster 2 (*n* = 476)		Cluster 3 (*n* = 220)	
T2DM cases	*n* = 145		*n* = 145			*n* = 25		*n* = 32		*n* = 88	
	OR	95% CI	OR	95% CI	p-interaction ^f^	OR	95% CI	OR	95% CI	OR	95% CI
**AHEI**											
Model 1 ^a^	**0.59**	**0.46–0.75**	**0.75**	**0.57–0.97**	**0.031**	1.47	0.82–2.62	0.71	0.47–1.18	**0.61**	**0.41–0.90**
Model 2 ^b^	**0.64**	**0.49–0.83**	0.79	0.60–1.04	**0.022**	1.52	0.83–2.78	0.83	0.49–1.41	**0.60**	**0.39–0.90**
Model 3 ^c^	0.87	0.66–1.13	0.87	0.65–1.15	**0.035**	1.73	0.93–3.23	0.95	0.55–1.63	**0.60**	**0.39–0.92**
**MDS**											
Model 1 ^a^	**0.66**	**0.52–0.83**	**0.72**	**0.55–0.94**	**0.038**	1.29	0.75–2.21	0.61	0.37–1.01	**0.59**	**0.39–0.90**
Model 2 ^b^	**0.74**	**0.57–0.96**	0.78	0.58–1.03	**0.029**	1.40	0.76–2.55	0.75	0.43–1.29	**0.62**	**0.40–0.96**
Model 3 ^c^	0.85	0.65–1.11	0.82	0.61–1.09	0.056	1.49	0.80–2.79	0.78	0.45–1.35	0.65	0.41–1.03

^a^: Adjusted for age, sex, energy intake. ^b^: Adjusted for age, sex, energy intake, education, physical activity, smoking. ^c^: Adjusted for age, sex, energy intake, education, physical activity, smoking, waist circumference, hypertension. ^d^: Further adjusted for metabotypes. ^e^: Given the low frequency of current smokers, logistic regressions in cluster 1 were adjusted for smoking status as never vs. former/current, ^f^: Interaction between metabotype and diet quality score. Mediterranean Diet Score (MDS): models per 2 points increase in the MDS score; Alternate Healthy Eating Index (AHEI): models per 10 points increase in the AHEI score. Significant associations are printed in bold.

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
