# Peer review of "Association of Dietary Patterns and Type-2 Diabetes Mellitus in Metabolically Homogeneous Subgroups in the KORA FF4 Study"

_nutrients, 2020, doi:10.3390/nu12061684_

Round 1
Reviewer 1 Report
The study by Wawro et al. evaluated the association of two dietary patterns and T2DM in three metabolically homogenous cohorts, stratified according to their metabotypes using 16 clinical markers.
The study is interesting and represents an attempt to better and more precisely profile such relationship and the risk of developing T2DM.
The dietary indices are significantly associatd with T2DM in the whole population.
Among the 3 subgroups, only the most unfavourable one was associated with T2DM.
Specific comments.
In the title I would define "diabetes" as "type 2 diabetes mellitus"
In the entire text "diabetes" should be defined as "diabetes mellitus".
Please specify ethnicity of the cohort.
The 3 metabotypes include a different proportion of males and females: please comment.
Author Response
We would like to thank all reviewers for their helpful comments that improved the manuscript. A native speaker checked and corrected the manuscript before we submitted the revised version of the manuscript. Line numbers given in the responses refer to the version with all markups visible.
Please find the detailed responses to each reviewer in the attachment.

Reviewer 2 Report
Association of dietary patterns and diabetes in metabolically homogenous subgroups in the KORA FF4 study
General comments
The manuscript outlined that Mediterranean Diet Score (MDS) and the Alternate Healthy Eating Index (AHEI) are inversely associated with the odds of type 2 diabetes mellitus, in a metabotype-dependent manner. The results are interesting, but not thoroughly interpreted and elaborated. Major and minor comments are as follows:
Major comments
- The paragraphs in the introduction do not have a concluding remark, which somehow makes the elaboration incomplete. For example, in Paragraph 1 (Lines 54-58), the authors highlighted some statistics about the prevalence of diabetes. However, without a concluding remark, the importance of mentioning these statistics is unclear. If the authors add in a sentence like “Hence, the widespread of diabetes clearly highlights the ever-increasing severity of the disease and the necessity to investigate its underlying pathophysiology so that better therapeutic strategy can be developed”, then the readers will be able capture the points that the authors try to deliver. A concluding sentence should be added to other paragraphs in the introduction.
- The description of the baseline characteristics is overly simplified. For example, Cluster 3 not only had the highest mean age, BMI and diabetes prevalence, but also had the lowest physically active and non-smoker population, highest hypertensive population, highest consumption of meat and meat products, and lowest consumption of vegetables, fruits and nuts. To my opinion, these differences should be highlighted so that the readers can understand the heterogeneity of between different clusters. In addition, if more detailed description of the study population has been published elsewhere, i.e. in Ref. 35 and 36, it should be cited in the result section.
- The description about the results of logistic regression models is somewhat lacking considering that it is the most crucial part of the result section in this manuscript. The statistical results were mentioned without adequate interpretations. For example, it was mentioned in line 211 that “significant associations between T2D and both scores were found in Models 1 and 2 for the total study sample”. This remark simply mentions the statistical significance, but the true meaning of the result, whereby the diet scores are negatively predictors of T2D odds, is not clearly specified. It is also unclear to the readers why the diet scores are no longer independent predictors of T2D odds in different models, partly because the rationales of using different covariances for adjustment of different models are not immediate apparent to the readers. Therefore, this paragraph should undergo a serious overhaul. In my opinion, based on Table 4, the authors should attempt to clarify the following matters:
- Differences of the diet scores as predictors of T2D odds (within and between each metabotype) in different models and briefly describe the implications (more details can be elaborated in Discussion)
- The extent of benefits predicted by the diet scores in the odds reduction. In Lines 218-220, it was mentioned that “OR of 0.60 indicates a lower odds of T2D for every 10 point increase in AHEI…”. In fact, an OR of 0.60 precisely means a reduction of T2D odds by 40%. I think the extent of reduction should be described to highlight the clinical relevance.
- Differences between the performance of AHEI and MDS scoring systems
- A large portion of the discussion focused on presenting the supporting evidence of association between the diet scores and T2D status, when the aim of the study was to “examine the associations of dietary patterns and T2D while accounting for metabotype.” Indeed, accomplishing the objective can help to develop tailored nutritional recommendations for individuals with different metabotypes. However, these interesting aspects are not well-discussed by the authors. In the results, we observed that only the T2D odds of Cluster 3 was inversely associated with the diet scores. Does this result indicate that Clusters 1 and 2 will not be benefited from changing their dietary compositions? What kind of nutritional advice can be derived from the result? These questions should be discussed and the clinical implications of the findings (if any) should be included in the abstract and conclusion.
Minor comments
- Line 69, defined as a priori and a useful …
- Line 71, …have been shown to be mostly consistently associated with…
- Line 132, … National Nutrient Database (Bundeslebensmittelschlüssel 3.02), the estimates of the…
- Line 134, the alignment of this paragraph was not justified.
- Line 140, For each food component, from 0 point to up to 10 points Based on the quantity of each food component consumed, 0 to 10 points were attributed, yielding a theoretical…
- Line 153, the alignment of Table 1 is not consistent.
- The association between the diet scores and T2D status remained significant even after adjustment with energy intake and physical activity. This information may imply that the type of food consumed remains as a predictor of T2D status independent of calorie deficit (PMID: 21197397). This is worth highlighting and discussed.
- Table 4, the subscripts of Models 1, 2 and 3 for MDS were incorrectly formatted.
- Table 4, the significant figure of OR for MDS model 2 (0.784) is inconsistent.
- Table 4, the sample sizes shown are the number of diabetic patients within the total study population or each cluster which is confusing. Instead, the actual sample sizes used to fit the statistical models should be presented. The number of diabetic patients can be presented in addition to the actual sample sizes used in each regression model.
- There are many other approaches to classify the metabotypes of an individual, i.e. anthropometric and biochemical parameters (PMID 22458475), gut microbiota and metabolomic profiles (PMID 31782487). Are there other studies which also demonstrated that dietary patterns could stratify the risk of T2D in a metabotype-dependent manner? When compared your metabotyping method, to other similar metabotyping approaches i.e. PMID 22458475, are there any differences in terms of the findings? I think these questions should be addressed in the discussion.
Author Response

(The authors gave the same response as above.)

Reviewer 3 Report
This manuscript describes a sub-analysis of the population-based KORA FF4 study that investigated the association between dietary patterns and type 2 diabetes. There are a considerable number of previous studies with similar focus, therefore this study is mainly confirmatory with minor degree of innovation based on the population stratification on metabotypes. The study results and conclusions derived from testing the hypothesis were along with what would have been expected. This being said, the study is still a valid and sound scientific contribution that consolidates previous knowledge. However, there are a considerable number of pitfalls that must be addressed by the authors:
1- The sentence in line 74-75 "it is well known that individual responses to dietary factors may vary due to variability in metabolic characteristics of individuals" should be rephrased and down stated as this is largely speculative and there is not solid evidence to support it. Moreover, using 5 references that are mostly opinion papers to support this statement is also inappropriate. If/whenever original research with the most solid evidence are available these should be used instead.
2- For those readers less acquainted with the principles of "metabotyping", a brief description of advantages and limitations should be included in the introduction and precede the sentence in line 75-78 " Consequently, identification of metabolotypes..." Once more, 10 references to support this general statement is excessive and several do not apply.
3- In the methods description, despite the fact that this is a sub-analysis of a larger study, a briefly more detailed description of the study methods would be welcome. What was the time frame of physical examinations and biochemical assessments, as compared to the dietary questionnaires. In particular in what concerns biochemical blood tests no information is provided in the manuscript.
4- More detail on how did the authors made the conversion of g/day intake into standard portion sizes, stated in line 148-149. Is there a standard methodology that was previously used, which can be referenced?
5- Table 4 needs to be improved for readability. In table 4, I can hardly identify the significant associations that the authors state to be in bold, as all figures seem the same. So I had to rely on the results description and follow the discussion to guess the significance that the authors had an intention to highlight in bold as written in the table legend.
Author Response

(The authors gave the same response as above.)

Reviewer 4 Report
Authors present an interesting study which allows them to determine if the metabotype influence the relationship between quality of diet and T2D in bavarian population. I think that is an valuable approach which can contribute to stablish an adequate personalized nutrition. I just have a few comments for the authors.
- Could the authors explain in the present manuscript what the criteria to choose the 16 clinical markers was follow to determine the metabotype? I know that details are given in other publication but I would be useful for readears if you also summarize shortly in the current one.
- Did the authors take into account the race of the population or it was not neccesary for this population?
- Table 4 is a bit confusing. The authors stated in the footnote that significant differences are in bold but nothing is in bold except headings. Maybe it was a format problem and the original table was correct, but you should check it.
- In the discussion, lines 232-238: could the authors explain a little bit more why they think that the introduction of waist circunference and hypertension eliminate associations between diet and T2M only for MDS? Do you have any idea/hypothesis of which kind of mediation effects could be? why did you think that the introduction could understimate the "true effect"?.
- There are some small format mistakes in table 1 and 4, and the paragraph in lines 134-152 is centered but not justified.
Author Response

(The authors gave the same response as above.)

Round 2
Reviewer 3 Report
The authors have adequately addressed all the comments and have made the requested alterations into the manuscript, which has improved significantly its readability and scientific accuracy.
Nevertheless, I still have some reservations concerning the sentence introduced in lines 291-293:
"This might be explained by the fact that these two factors are potentially in the causal pathway, i.e. potential mediators, of diet and T2DM association. In fact, a low diet quality could lead to a high waist circumference and/or hypertension, which in turn could lead to T2DM."
In particular the wording "causal pathway" and "could lead to" should be rephrased and down graded, since there is no sufficient evidence from the authors work or others to support such a causality/direct relationship between diet and waist circumference, hypertension nor diabetes, which aetiologies are recognized to be multifactorial.
Additionally, despite the authors revised the reference list by removing redundant citations, greater efforts could have been made.
Author Response
We thank the reviewer for the critical remarks.
We changed the manuscript (lines 270-276) to:
When additionally including waist circumference and hypertension as adjustment variables in the logistic regression models, the association between T2DM and diet quality scores was attenuated. This might be explained by the fact that these two factors are potential mediators of diet and T2DM association (e.g. [16,44-46]). It is known that mediating factors should not be included as adjusting factors in regression models. Therefore, our Model 3 was potentially overadjusted. The cross-sectional design of the KORA FF4 study does, however, not allow for examining the potential mediation effect of the above-mentioned variables.
During the first revision, we removed redundant references. A further reduction of references on the issue of metabotyping, however, would lead to an unbalanced statement. Thus, we would like to keep the currently cited references.
